# When to Trust Your Model: Model-Based Policy Optimization

**Michael Janner**     **Justin Fu**     **Marvin Zhang**     **Sergey Levine**
University of California, Berkeley
{janner, justinjfu, marvin, svlevine}@eecs.berkeley.edu

## Abstract

Designing effective model-based reinforcement learning algorithms is difficult because the ease of data generation must be weighed against the bias of model-generated data. In this paper, we study the role of model usage in policy optimization both theoretically and empirically. We first formulate and analyze a model-based reinforcement learning algorithm with a guarantee of monotonic improvement at each step. In practice, this analysis is overly pessimistic and suggests that real off-policy data is always preferable to model-generated on-policy data, but we show that an empirical estimate of model generalization can be incorporated into such analysis to justify model usage. Motivated by this analysis, we then demonstrate that a simple procedure of using short model-generated rollouts branched from real data has the benefits of more complicated model-based algorithms without the usual pitfalls. In particular, this approach surpasses the sample efficiency of prior model-based methods, matches the asymptotic performance of the best model-free algorithms, and scales to horizons that cause other model-based methods to fail entirely.

## 1   Introduction

Reinforcement learning algorithms generally fall into one of two categories: model-based approaches, which build a predictive model of an environment and derive a controller from it, and model-free techniques, which learn a direct mapping from states to actions. Model-free methods have shown promise as a general-purpose tool for learning complex policies from raw state inputs (Mnih et al., 2015; Lillicrap et al., 2016; Haarnoja et al., 2018), but their generality comes at the cost of efficiency. When dealing with real-world physical systems, for which data collection can be an arduous process, model-based approaches are appealing due to their comparatively fast learning. However, model accuracy acts as a bottleneck to policy quality, often causing model-based approaches to perform worse asymptotically than their model-free counterparts.

In this paper, we study how to most effectively use a predictive model for policy optimization. We first formulate and analyze a class of model-based reinforcement learning algorithms with improvement guarantees. Although there has been recent interest in monotonic improvement of model-based reinforcement learning algorithms (Sun et al., 2018; Luo et al., 2019), most commonly used model-based approaches lack the improvement guarantees that underpin many model-free methods (Schulman et al., 2015). While it is possible to apply analogous techniques to the study of model-based methods to achieve similar guarantees, it is more difficult to use such analysis to justify model usage in the first place due to pessimistic bounds on model error. However, we show that more realistic model error rates derived empirically allow us to modify this analysis to provide a more reasonable tradeoff on model usage.

Our main contribution is a practical algorithm built on these insights, which we call model-based policy optimization (MBPO), that makes limited use of a predictive model to achieve pronounced

improvements in performance compared to other model-based approaches. More specifically, we disentangle the task horizon and model horizon by querying the model only for short rollouts. We empirically demonstrate that a large amount of these short model-generated rollouts can allow a policy optimization algorithm to learn substantially faster than recent model-based alternatives while retaining the asymptotic performance of the most competitive model-free algorithms. We also show that MBPO does not suffer from the same pitfalls as prior model-based approaches, avoiding model exploitation and failure on long-horizon tasks. Finally, we empirically investigate different strategies for model usage, supporting the conclusion that careful use of short model-based rollouts provides the most benefit to a reinforcement learning algorithm.

## 2 Related work

Model-based reinforcement learning methods are promising candidates for real-world sequential decision-making problems due to their data efficiency (Kaelbling et al., 1996). Gaussian processes and time-varying linear dynamical systems provide excellent performance in the low-data regime (Deisenroth & Rasmussen, 2011; Levine & Koltun, 2013; Kumar et al., 2016). Neural network predictive models (Draeger et al., 1995; Gal et al., 2016; Depeweg et al., 2016; Nagabandi et al., 2018), are appealing because they allow for algorithms that combine the sample efficiency of a model-based approach with the asymptotic performance of high-capacity function approximators, even in domains with high-dimensional observations (Oh et al., 2015; Ebert et al., 2018; Kaiser et al., 2019). Our work uses an ensemble of probabilistic networks, as in Chua et al. (2018), although our model is employed to learn a policy rather than in the context of a receding-horizon planning routine.

Learned models may be incorporated into otherwise model-free methods for improvements in data efficiency. For example, a model-free policy can be used as an action proposal distribution within a model-based planner (Piché et al., 2019). Conversely, model rollouts may be used to provide extra training examples for a Q-function (Sutton, 1990), to improve the target value estimates of existing data points (Feinberg et al., 2018), or to provide additional context to a policy (Du & Narasimhan, 2019). However, the performance of such approaches rapidly degrades with increasing model error (Gu et al., 2016), motivating work that interpolates between different rollout lengths (Buckman et al., 2018), tunes the ratio of real to model-generated data (Kalweit & Boedecker, 2017), or does not rely on model predictions (Heess et al., 2015). Our approach similarly tunes model usage during policy optimization, but we show that justifying non-negligible model usage during most points in training requires consideration of the model's ability to generalize outside of its training distribution.

Prior methods have also explored incorporating computation that resembles model-based planning but without constraining the intermediate predictions of the planner to match plausible environment observations (Tamar et al., 2016; Racanière et al., 2017; Oh et al., 2017; Silver et al., 2017). While such methods can reach asymptotic performance on par with model-free approaches, they may not benefit from the sample efficiency of model-based methods as they forgo the extra supervision used in standard model-based methods.

The bottleneck in scaling model-based approaches to complex tasks often lies in learning reliable predictive models of high-dimensional dynamics (Atkeson & Schaal, 1997). While ground-truth models are most effective when queried for long horizons (Holland et al., 2018), inaccuracies in learned models tend to make long rollouts unreliable. Ensembles have shown to be effective in preventing a policy or planning procedure from exploiting such inaccuracies (Rajeswaran et al., 2017; Kurutach et al., 2018; Clavera et al., 2018; Chua et al., 2018). Alternatively, a model may also be trained on its own outputs to avoid compounding error from multi-step predictions (Talvitie, 2014, 2016) or predict many timesteps into the future (Whitney & Fergus, 2018). We demonstrate that a combination of model ensembles with short model rollouts is sufficient to prevent model exploitation.

Theoretical analysis of model-based reinforcement learning algorithms has been considered by Sun et al. (2018) and Luo et al. (2019), who bound the discrepancy between returns under a model and those in the real environment of interest. Their approaches enforce a trust region around a reference policy, whereas we do not constrain the policy but instead consider rollout length based on estimated model generalization capacity. Alternate analyses have been carried out by incorporating the structure of the value function into the model learning (Farahmand et al., 2017) or by regularizing the model by controlling its Lipschitz constant (Asadi et al., 2018). Prior work has also constructed complexity bounds for model-based approaches in the tabular setting (Szita & Szepesvari, 2010) and for the linear quadratic regulator (Dean et al., 2017), whereas we consider general non-linear systems.

---

**Algorithm 1** Monotonic Model-Based Policy Optimization

---

1: Initialize policy $\pi(a|s)$, predictive model $p_\theta(s', r|s, a)$, empty dataset $\mathcal{D}$.
2: **for** $N$ epochs **do**
3:     Collect data with $\pi$ in real environment: $\mathcal{D} = \mathcal{D} \cup \{(s_i, a_i, s'_i, r_i)\}_i$
4:     Train model $p_\theta$ on dataset $\mathcal{D}$ via maximum likelihood: $\theta \leftarrow \text{argmax}_\theta \mathbb{E}_\mathcal{D}[\log p_\theta(s', r|s, a)]$
5:     Optimize policy under predictive model: $\pi \leftarrow \text{argmax}_{\pi'} \hat{\eta}[\pi'] - C(\epsilon_m, \epsilon_\pi)$

---

## 3 Background

We consider a Markov decision process (MDP), defined by the tuple $(\mathcal{S}, \mathcal{A}, p, r, \gamma, \rho_0)$. $\mathcal{S}$ and $\mathcal{A}$ are the state and action spaces, respectively, and $\gamma \in (0, 1)$ is the discount factor. The dynamics or transition distribution are denoted as $p(s'|s, a)$, the initial state distribution as $\rho_0(s)$, and the reward function as $r(s, a)$. The goal of reinforcement learning is to find the optimal policy $\pi^*$ that maximizes the expected sum of discounted rewards, denoted by $\eta$:

$$\pi^* = \operatorname*{argmax}_\pi \eta[\pi] = \operatorname*{argmax}_\pi E_\pi \left[ \sum_{t=0}^\infty \gamma^t r(s_t, a_t) \right].$$

The dynamics $p(s'|s, a)$ are assumed to be unknown. Model-based reinforcement learning methods aim to construct a model of the transition distribution, $p_\theta(s'|s, a)$, using data collected from interaction with the MDP, typically using supervised learning. We additionally assume that the reward function has unknown form, and predict $r$ as a learned function of $s$ and $a$.

## 4 Monotonic improvement with model bias

In this section, we first lay out a general recipe for MBPO with monotonic improvement. This general recipe resembles or subsumes several prior algorithms and provides us with a concrete framework that is amenable to theoretical analysis. Described generically in Algorithm 1, MBPO optimizes a policy under a learned model, collects data under the updated policy, and uses that data to train a new model. While conceptually simple, the performance of MBPO can be difficult to understand; errors in the model can be exploited during policy optimization, resulting in large discrepancies between the predicted returns of the policy under the model and under the true dynamics.

### 4.1 Monotonic model-based improvement

Our goal is to outline a principled framework in which we can provide performance guarantees for model-based algorithms. To show monotonic improvement for a model-based method, we wish to construct a bound of the following form:

$$\eta[\pi] \geq \hat{\eta}[\pi] - C.$$

$\eta[\pi]$ denotes the returns of the policy in the true MDP, whereas $\hat{\eta}[\pi]$ denotes the returns of the policy under our model. Such a statement guarantees that, as long as we improve by at least $C$ under the model, we can guarantee improvement on the true MDP.

The gap between true returns and model returns, $C$, can be expressed in terms of two error quantities of the model: generalization error due to sampling, and distribution shift due to the updated policy encountering states not seen during model training. As the model is trained with supervised learning, sample error can be quantified by standard PAC generalization bounds, which bound the difference in expected loss and empirical loss by a constant with high probability (Shalev-Shwartz & Ben-David, 2014). We denote this generalization error by $\epsilon_m = \max_t E_{s \sim \pi_{D,t}}[D_{TV}(p(s', r|s, a)||p_\theta(s', r|s, a))]$, which can be estimated in practice by measuring the validation loss of the model on the time-dependent state distribution of the data-collecting policy $\pi_D$. For our analysis, we denote distribution shift by the maximum total-variation distance, $\max_s D_{TV}(\pi||\pi_D) \leq \epsilon_\pi$, of the policy between iterations. In practice, we measure the KL divergence between policies, which we can relate to $\epsilon_\pi$ by Pinsker's inequality. With these two sources of error controlled (generalization by $\epsilon_m$, and distribution shift by $\epsilon_\pi$), we now present our bound:

**Theorem 4.1.** *Let the expected TV-distance between two transition distributions be bounded at each timestep by $\epsilon_m$ and the policy divergence be bounded by $\epsilon_\pi$. Then the true returns and model returns of the policy are bounded as:*

$$\eta[\pi] \geq \hat{\eta}[\pi] - \underbrace{\left[ \frac{2\gamma r_{\max}(\epsilon_m + 2\epsilon_\pi)}{(1-\gamma)^2} + \frac{4r_{\max}\epsilon_\pi}{(1-\gamma)} \right]}_{C(\epsilon_m, \epsilon_\pi)} \tag{1}$$

*Proof.* See Appendix A, Theorem A.1. □

This bound implies that as long as we improve the returns under the model $\hat{\eta}[\pi]$ by more than $C(\epsilon_m, \epsilon_\pi)$, we can guarantee improvement under the true returns.

## 4.2 Interpolating model-based and model-free updates

Theorem 4.1 provides a useful relationship between model returns and true returns. However, it contains several issues regarding cases when the model error $\epsilon_m$ is high. First, there may not exist a policy such that $\hat{\eta}[\pi] - \eta[\pi] > C(\epsilon_m, \epsilon_\pi)$, in which case improvement is not guaranteed. Second, the analysis relies on running full rollouts through the model, allowing model errors to compound. This is reflected in the bound by a factor scaling quadratically with the effective horizon, $1/(1-\gamma)$. In such cases, we can improve the algorithm by choosing to rely less on the model and instead more on real data collected from the true dynamics when the model is inaccurate.

In order to allow for dynamic adjustment between model-based and model-free rollouts, we introduce the notion of a *branched rollout*, in which we begin a rollout from a state under the previous policy's state distribution $d_{\pi_D}(s)$ and run $k$ steps according to $\pi$ under the learned model $p_\theta$. This branched rollout structure resembles the scheme proposed in the original Dyna algorithm (Sutton, 1990), which can be viewed as a special case of a length 1 branched rollouts. Formally, we can view this as executing a nonstationary policy which begins a rollout by sampling actions from the previous policy $\pi_D$. Then, at some specified time, we switch to unrolling the trajectory under the model $p$ and current policy $\pi$ for $k$ steps. Under such a scheme, the returns can be bounded as follows:

**Theorem 4.2.** *Given returns $\eta^{\text{branch}}[\pi]$ from the $k$-branched rollout method,*

$$\eta[\pi] \geq \eta^{\text{branch}}[\pi] - 2r_{\max} \left[ \frac{\gamma^{k+1}\epsilon_\pi}{(1-\gamma)^2} + \frac{\gamma^k + 2}{(1-\gamma)}\epsilon_\pi + \frac{k}{1-\gamma}(\epsilon_m + 2\epsilon_\pi) \right]. \tag{2}$$

*Proof.* See Appendix A, Theorem A.3. □

## 4.3 Model generalization in practice

Theorem 4.2 would be most useful for guiding algorithm design if it could be used to determine an optimal model rollout length $k$. While this bound does include two competing factors, one exponentially decreasing in $k$ and another scaling linearly with $k$, the values of the associated constants prevent an actual tradeoff; taken literally, this lower bound is maximized when $k = 0$, corresponding to not using the model at all. One limitation of the analysis is pessimistic scaling of model error $\epsilon_m$ with respect to policy shift $\epsilon_\pi$, as we do not make any assumptions about the generalization capacity or smoothness properties of the model (Asadi et al., 2018).

To better determine how well we can expect our model to generalize in practice, we empirically measure how the model error under new policies increases with policy change $\epsilon_\pi$. We train a model on the state distribution of a data-collecting policy $\pi_D$ and then continue policy optimization while measuring the model's loss on all intermediate policies $\pi$ during this optimization. Figure 1a shows that, as expected, the model error increases with the divergence between the current policy $\pi$ and the data-collecting policy $\pi_D$. However, the rate of this increase depends on the amount of data collected by $\pi_D$. We plot the local change in model error over policy change, $\frac{d\epsilon_{m'}}{d\epsilon_\pi}$, in Figure 1b. The decreasing dependence on policy shift shows that not only do models trained with more data perform better on their training distribution, but they also generalize better to nearby distributions.

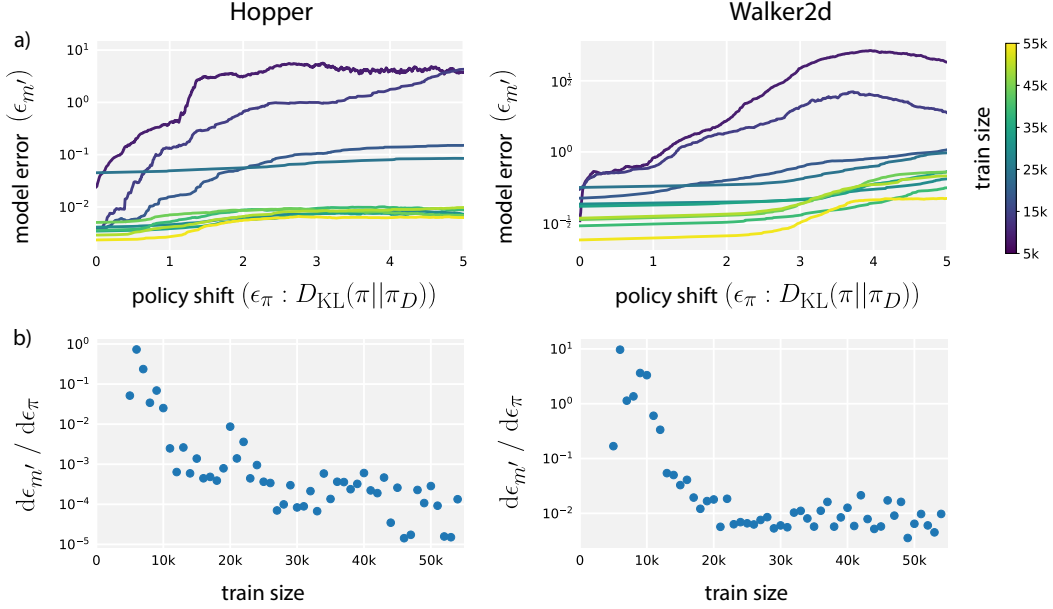

Figure 1: (a) We train a predictive model on the state distribution of $\pi_D$ and evaluate it on policies $\pi$ of varying KL-divergence from $\pi_D$ without retraining. The color of each curve denotes the amount of data from $\pi_D$ used to train the model corresponding to that curve. The offsets of the curves depict the expected trend of increasing training data leading to decreasing model error on the training distribution. However, we also see a decreasing influence of state distribution shift on model error with increasing training data, signifying that the model is generalizing better. (b) We measure the local change in model error versus KL-divergence of the policies at $\epsilon_\pi = 0$ as a proxy to model generalization.

The clear trend in model error growth rate suggests a way to modify the pessimistic bounds. In the previous analysis, we assumed access to only model error $\epsilon_m$ on the distribution of the most recent data-collecting policy $\pi_D$ and approximated the error on the current distribution as $\epsilon_m + 2\epsilon_\pi$. If we can instead approximate the model error on the distribution of the current policy $\pi$, which we denote as $\epsilon_{m'}$, we may use this directly. For example, approximating $\epsilon_{m'}$ with a linear function of the policy divergence yields:

$$\hat{\epsilon}_{m'}(\epsilon_\pi) \approx \epsilon_m + \epsilon_\pi \frac{\mathrm{d}\epsilon_{m'}}{\mathrm{d}\epsilon_\pi}$$

where $\frac{\mathrm{d}\epsilon_{m'}}{\mathrm{d}\epsilon_\pi}$ is empirically estimated as in Figure 1. Equipped with an approximation of $\epsilon_{m'}$, the model's error on the distribution of the current policy $\pi$, we arrive at the following bound:

**Theorem 4.3.** *Under the $k$-branched rollout method, using model error under the updated policy* $\epsilon_{m'} \geq \max_t E_{s \sim \pi_{D,t}}[D_{TV}(p(s'|s,a)||\hat{p}(s'|s,a))]$, *we have*

$$\eta[\pi] \geq \eta^{\mathrm{branch}}[\pi] - 2r_{\max}\left[\frac{\gamma^{k+1}\epsilon_\pi}{(1-\gamma)^2} + \frac{\gamma^k \epsilon_\pi}{(1-\gamma)} + \frac{k}{1-\gamma}(\epsilon_{m'})\right]. \tag{3}$$

*Proof.* See Appendix A, Theorem A.2. $\qquad\qquad\qquad\qquad\qquad\qquad\qquad\qquad\qquad\qquad\qquad\square$

While this bound appears similar to Theorem 4.2, the important difference is that this version actually motivates model usage. More specifically, $k^* = \underset{k}{\mathrm{argmin}}\left[\frac{\gamma^{k+1}\epsilon_\pi}{(1-\gamma)^2} + \frac{\gamma^k \epsilon_\pi}{(1-\gamma)} + \frac{k}{1-\gamma}(\epsilon_{m'})\right] > 0$ for sufficiently low $\epsilon_{m'}$. While this insight does not immediately suggest an algorithm design by itself, we can build on this idea to develop a method that makes limited use of truncated, but nonzero-length, model rollouts.

---

**Algorithm 2** Model-Based Policy Optimization with Deep Reinforcement Learning
---
1: Initialize policy $\pi_\phi$, predictive model $p_\theta$, environment dataset $\mathcal{D}_{\text{env}}$, model dataset $\mathcal{D}_{\text{model}}$
2: **for** $N$ epochs **do**
3:     Train model $p_\theta$ on $\mathcal{D}_{\text{env}}$ via maximum likelihood
4:     **for** $E$ steps **do**
5:         Take action in environment according to $\pi_\phi$; add to $\mathcal{D}_{\text{env}}$
6:         **for** $M$ model rollouts **do**
7:             Sample $s_t$ uniformly from $\mathcal{D}_{\text{env}}$
8:             Perform $k$-step model rollout starting from $s_t$ using policy $\pi_\phi$; add to $\mathcal{D}_{\text{model}}$
9:         **for** $G$ gradient updates **do**
10:            Update policy parameters on model data: $\phi \leftarrow \phi - \lambda_\pi \hat{\nabla}_\phi J_\pi(\phi, \mathcal{D}_{\text{model}})$
---

## 5  Model-based policy optimization with deep reinforcement learning

We now present a practical model-based reinforcement learning algorithm based on the derivation in the previous section. Instantiating Algorithm 1 amounts to specifying three design decisions: (1) the parametrization of the model $p_\theta$, (2) how the policy $\pi$ is optimized given model samples, and (3) how to query the model for samples for policy optimization.

**Predictive model.**  In our work, we use a bootstrap ensemble of dynamics models $\{p_\theta^1, ..., p_\theta^B\}$. Each member of the ensemble is a probabilistic neural network whose outputs parametrize a Gaussian distribution with diagonal covariance: $p_\theta^i(s_{t+1}, r|s_t, a_t) = \mathcal{N}(\mu_\theta^i(s_t, a_t), \Sigma_\theta^i(s_t, a_t)))$. Individual probabilistic models capture aleatoric uncertainty, or the noise in the outputs with respect to the inputs. The bootstrapping procedure accounts for epistemic uncertainty, or uncertainty in the model parameters, which is crucial in regions when data is scarce and the model can by exploited by policy optimization. Chua et al. (2018) demonstrate that a proper handling of both of these uncertainties allows for asymptotically competitive model-based learning. To generate a prediction from the ensemble, we simply select a model uniformly at random, allowing for different transitions along a single model rollout to be sampled from different dynamics models.

**Policy optimization.**  We adopt soft-actor critic (SAC) (Haarnoja et al., 2018) as our policy optimization algorithm. SAC alternates between a policy evaluation step, which estimates $Q^\pi(s, a) = E_\pi[\sum_{t=0}^\infty \gamma^t r(s_t, a_t)|s_0 = s, a_0 = a]$ using the Bellman backup operator, and a policy improvement step, which trains an actor $\pi$ by minimizing the expected KL-divergence $J_\pi(\phi, \mathcal{D}) = \mathbb{E}_{s_t \sim \mathcal{D}}[D_{KL}(\pi|| \exp\{Q^\pi - V^\pi\})]$.

**Model usage.**  Many recent model-based algorithms have focused on the setting in which model rollouts begin from the initial state distribution (Kurutach et al., 2018; Clavera et al., 2018). While this may be a more faithful interpretation of Algorithm 1, as it is optimizing a policy purely under the state distribution of the model, this approach entangles the model rollout length with the task horizon. Because compounding model errors make extended rollouts difficult, these works evaluate on truncated versions of benchmarks. The branching strategy described in Section 4.2, in which model rollouts begin from the state distribution of a different policy under the true environment dynamics, effectively relieves this limitation. In practice, branching replaces few long rollouts from the initial state distribution with many short rollouts starting from replay buffer states.

A practical implementation of MBPO is described in Algorithm 2.[1] The primary differences from the general formulation in Algorithm 1 are $k$-length rollouts from replay buffer states in the place of optimization under the model's state distribution and a fixed number of policy update steps in the place of an intractable argmax. Even when the horizon length $k$ is short, we can perform many such short rollouts to yield a large set of model samples for policy optimization. This large set allows us to take many more policy gradient steps per environment sample (between 20 and 40) than is typically stable in model-free algorithms. A full listing of the hyperparameters included in Algorithm 2 for all evaluation environments is given in Appendix C.

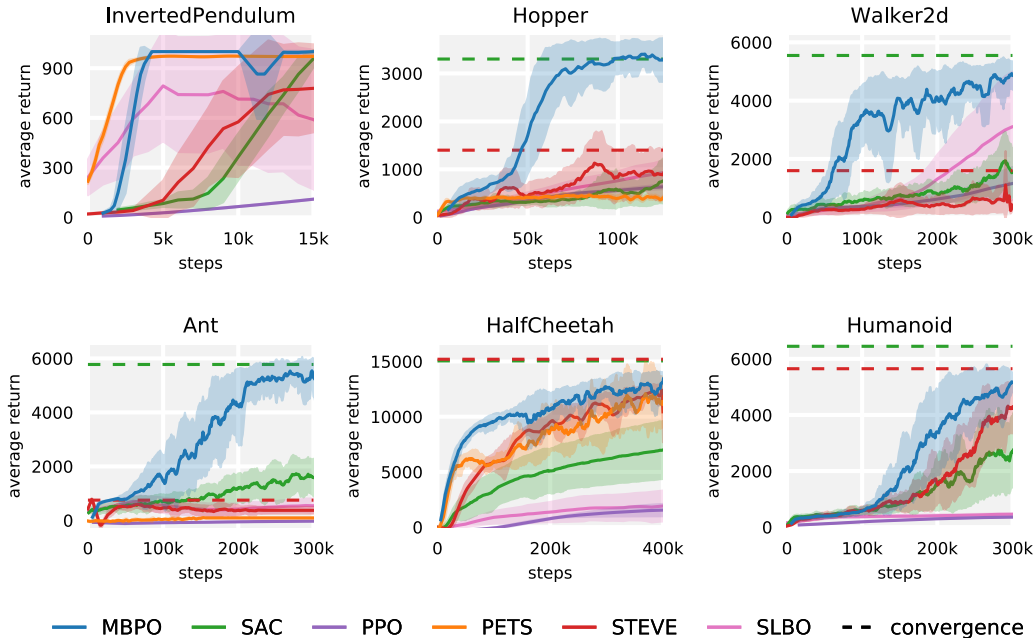

Figure 2: Training curves of MBPO and five baselines on continuous control benchmarks. Solid curves depict the mean of five trials and shaded regions correspond to standard deviation among trials. MBPO has asymptotic performance similar to the best model-free algorithms while being faster than the model-based baselines. For example, MBPO's performance on the Ant task at 300 thousand steps matches that of SAC at 3 million steps. We evaluated all algorithms on the standard 1000-step versions of the benchmarks.

## 6 Experiments

Our experimental evaluation aims to study two primary questions: (1) How well does MBPO perform on benchmark reinforcement learning tasks, compared to state-of-the-art model-based and model-free algorithms? (2) What conclusions can we draw about appropriate model usage?

### 6.1 Comparative evaluation

In our comparisons, we aim to understand both how well our method compares to state-of-the-art model-based and model-free methods and how our design choices affect performance. We compare to two state-of-the-art model-free methods, SAC (Haarnoja et al., 2018) and PPO (Schulman et al., 2017), both to establish a baseline and, in the case of SAC, measure the benefit of incorporating a model, as our model-based method uses SAC for policy learning as well. For model-based methods, we compare to PETS (Chua et al., 2018), which does not perform explicit policy learning, but directly uses the model for planning; STEVE (Buckman et al., 2018), which also uses short-horizon model-based rollouts, but incorporates data from these rollouts into value estimation rather than policy learning; and SLBO (Luo et al., 2019), a model-based algorithm with performance guarantees that performs model rollouts from the initial state distribution. These comparisons represent the state-of-the-art in both model-free and model-based reinforcement learning.

We evaluate MBPO and these baselines on a set of MuJoCo continuous control tasks (Todorov et al., 2012) commonly used to evaluate model-free algorithms. Note that some recent works in model-based reinforcement learning have used modified versions of these benchmarks, where the task horizon is chosen to be shorter so as to simplify the modeling problem (Kurutach et al., 2018; Clavera et al., 2018). We use the standard full-length version of these tasks. MBPO also does not assume access to privileged information in the form of fully observable states or the reward function for offline evaluation.

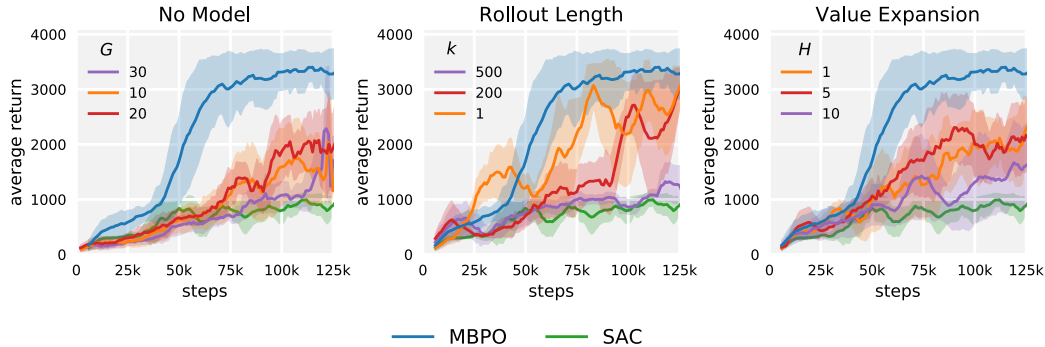

Figure 3: **No model:** SAC run without model data but with the same range of gradient updates per environment step ($G$) as MBPO on the Hopper task. **Rollout length:** While we find that increasing rollout length $k$ over time yields the best performance for MBPO (Appendix C), single-step rollouts provide a baseline that is difficult to beat. **Value expansion:** We implement $H$-step model value expansion from Feinberg et al. (2018) on top of SAC for a more informative comparison. We also find in the context of value expansion that single-step model rollouts are surprisingly competitive.

Figure 2 shows the learning curves for all methods, along with asymptotic performance of algorithms which do not converge in the region shown. These results show that MBPO learns substantially faster, an order of magnitude faster on some tasks, than prior model-free methods, while attaining comparable final performance. For example, MBPO's performance on the Ant task at 300 thousand steps is the same as that of SAC at 3 million steps. On Hopper and Walker2d, MBPO requires the equivalent of 14 and 40 minutes, respectively, of simulation time if the simulator were running in real time. More crucially, MBPO learns on some of the higher-dimensional tasks, such as Ant, which pose problems for purely model-based approaches such as PETS.

## 6.2 Design evaluation

We next make ablations and modifications to our method to better understand why MBPO outperforms prior approaches. Results for the following experiments are shown in Figure 3.

**No model.**   The ratio between the number of gradient updates and environment samples, $G$, is much higher in MBPO than in comparable model-free algorithms because the model-generated data reduces the risk of overfitting. We run standard SAC with similarly high ratios, but without model data, to ensure that the model is actually helpful. While increasing the number of gradient updates per sample taken in SAC does marginally speed up learning, we cannot match the sample-efficiency of our method without using the model. For hyperparameter settings of MBPO, see Appendix C.

**Rollout horizon.**   While the best-performing rollout length schedule on the Hopper task linearly increases from $k = 1$ to $15$ (Appendix C), we find that fixing the rollout length at 1 for the duration of training retains much of the benefit of our model-based method. We also find that our model is accurate enough for 200-step rollouts, although this performs worse than shorter values when used for policy optimization. 500-step rollouts are too inaccurate for effective learning. While more precise fine-tuning is always possible, augmenting policy training data with single-step model rollouts provides a baseline that is surprisingly difficult to beat and outperforms recent methods which perform longer rollouts from the initial state distribution. This result agrees with our theoretical analysis which prescribes short model-based rollouts to mitigate compounding modeling errors.

**Value expansion.**   An alternative to using model rollouts for direct training of a policy is to improve the quality of target values of samples collected from the real environment. This technique is used in model-based value expansion (MVE) (Feinberg et al., 2018) and STEVE (Buckman et al., 2018). While MBPO outperforms both of these approaches, there are other confounding factors making a head-to-head comparison difficult, such as the choice of policy learning algorithm. To better determine the relationship between training on model-generated data and using model predictions to

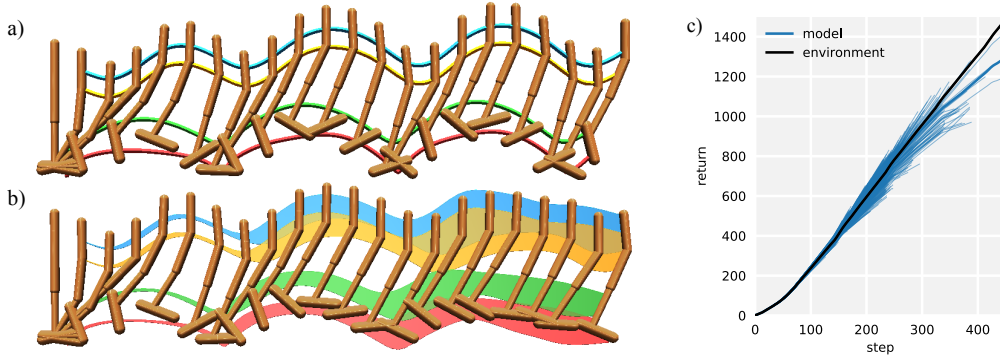

Figure 4: a) A 450-step hopping sequence performed in the real environment, with the trajectory of the body's joints traced through space. b) The same action sequence rolled out under the model 1000 times, with shaded regions corresponding to one standard deviation away from the mean prediction. The growing uncertainty and deterioration of a recognizable sinusoidal motion underscore accumulation of model errors. c) Cumulative returns of the same policy under the model and actual environment dynamics reveal that the policy is not exploiting the learned model. Thin blue lines reflect individual model rollouts and the thick blue line is their mean.

improve target values, we augment SAC with the $H$-step $Q$-target objective:

$$\frac{1}{H} \sum_{t=-1}^{H-1} (Q(\hat{s}_t, \hat{a}_t) - (\sum_{k=t}^{H-1} \gamma^{k-t} \hat{r}_k + \gamma^H Q(\hat{s}_H, \hat{a}_H)))^2$$

in which $\hat{s}_t$ and $\hat{r}_t$ are model predictions and $\hat{a}_t \sim \pi(a_t|\hat{s}_t)$. We refer the reader to Feinberg et al. (2018) for further discussion of this approach. We verify that SAC also benefits from improved target values, and similar to our conclusions from MBPO, single-step model rollouts ($H = 1$) provide a surprisingly effective baseline. While model-generated data augmentation and value expansion are in principle complementary approaches, preliminary experiments did not show improvements to MBPO by using improved target value estimates.

**Model exploitation.** We analyze the problem of "model exploitation," which a number of recent works have raised as a primary challenge in model-based reinforcement learning (Rajeswaran et al., 2017; Clavera et al., 2018; Kurutach et al., 2018). We plot empirical returns of a trained policy on the Hopper task under both the real environment and the model in Figure 4 (c) and find, surprisingly, that they are highly correlated, indicating that a policy trained on model-predicted transitions may not exploit the model at all if the rollouts are sufficiently short. This is likely because short rollouts are more likely to reflect the real dynamics (Figure 4 a-b), reducing the opportunities for policies to rely on inaccuracies of model predictions. While the models for other environments are not necessarily as accurate as that for Hopper, we find across the board that model returns tend to *underestimate* real environment returns in MBPO.

## 7   Discussion

We have investigated the role of model usage in policy optimization procedures through both a theoretical and empirical lens. We have shown that, while it is possible to formulate model-based reinforcement learning algorithms with monotonic improvement guarantees, such an analysis cannot necessarily be used to motivate using a model in the first place. However, an empirical study of model generalization shows that predictive models can indeed perform well outside of their training distribution. Incorporating a linear approximation of model generalization into the analysis gives rise to a more reasonable tradeoff that does in fact justify using the model for truncated rollouts. The algorithm stemming from this insight, MBPO, has asymptotic performance rivaling the best model-free algorithms, learns substantially faster than prior model-free or model-based methods, and scales to long horizon tasks that often cause model-based methods to fail. We experimentally investigate the tradeoffs associated with our design decisions, and find that model rollouts as short as a single step can provide pronounced benefits to policy optimization procedures.

**Acknowledgements**

We thank Anusha Nagabandi, Michael Chang, Chelsea Finn, Pulkit Agrawal, and Jacob Steinhardt for insightful discussions; Vitchyr Pong, Alex Lee, Kyle Hsu, and Aviral Kumar for feedback on an early draft of the paper; and Kristian Hartikainen for help with the SAC baseline. This research was partly supported by the NSF via IIS-1651843, IIS-1700697, and IIS-1700696, the Office of Naval Research, ARL DCIST CRA W911NF-17-2-0181, and computational resource donations from Google. M.J. is supported by fellowships from the National Science Foundation and the Open Philanthropy Project. M.Z. is supported by an NDSEG fellowship.

## Footnotes

[1] When SAC is used as the policy optimization algorithm, we must also perform gradient updates on the parameters of the $Q$-functions, but we omit these updates for clarity.

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
