[Supplementary Material]

# Appendices

## A   Model-based Policy Optimization with Performance Guarantees

In this appendix, we provide proofs for bounds presented in the main paper.

We begin with a standard bound on model-based policy optimization, with bounded policy change $\epsilon_\pi$ and model error $\epsilon_m$.

**Theorem A.1** (MBPO performance bound). *Let the expected total variation between two transition distributions is bounded at each timestep by $\max_t E_{s \sim \pi_{D,t}}[D_{TV}(p(s'|s,a)||\hat{p}(s'|s,a))] \leq \epsilon_m$, and the policy divergences are bounded as $\max_s D_{TV}(\pi_D(a|s)||\pi(a|s)) \leq \epsilon_\pi$. Then the returns are bounded as:*

$$\eta[\pi] \geq \hat{\eta}[\pi] - \frac{2\gamma r_{\max}(\epsilon_m + 2\epsilon_\pi)}{(1-\gamma)^2} - \frac{4r_{\max}\epsilon_\pi}{(1-\gamma)}$$

*Proof.* Let $\pi_D$ denote the data collecting policy. As-is we can use Lemma B.3 to bound the returns, but it will require bounded model error under the new policy $\pi$. Thus, we need to introduce $\pi_D$ by adding and subtracting $\eta[\pi_D]$, to get:

$$\eta[\pi] - \hat{\eta}[\pi] = \underbrace{\eta[\pi] - \eta[\pi_D]}_{L_1} + \underbrace{\eta[\pi_D] - \hat{\eta}[\pi]}_{L_2}$$

We can bound $L_1$ and $L_2$ both using Lemma B.3.

For $L_1$, we apply Lemma B.3 using $\delta = \epsilon_\pi$ (no model error because both terms are under the true model), and obtain:

$$L_1 \geq -\frac{2r_{\max}\gamma\epsilon_\pi}{(1-\gamma)^2} - \frac{2r_{\max}\epsilon_\pi}{1-\gamma}$$

For $L_1$, we apply Lemma B.3 using $\delta = \epsilon_\pi + \epsilon_m$ and obtain:

$$L_2 \geq -\frac{2r_{\max}\gamma(\epsilon_m + \epsilon_\pi)}{(1-\gamma)^2} - \frac{2r_{\max}\epsilon_\pi}{1-\gamma}$$

Adding these two bounds together yields the desired result. $\qquad\square$

Next, we describe bounds for branched rollouts. We define a branched rollout as a rollout which begins under some policy and dynamics (either true or learned), and at some point in time switches to rolling out under a new policy and dynamics for $k$ steps. The point at which the branch is selected is weighted exponentially in time – that is, the probability of a branch point $t$ being selected is proportional to $\gamma^t$.

We first present the simpler bound where the model error is bounded under the new policy, which we label as $\epsilon_{m'}$. This bound is difficult to apply in practice as supervised learning will typically control model error under the dataset collected by the previous policy.

**Theorem A.2.** *Let the expected total variation between two the learned model is bounded at each timestep under the expectation of $\pi$ by $\max_t E_{s \sim \pi_t}[D_{TV}(p(s'|s,a)||\hat{p}(s'|s,a))] \leq \epsilon_{m'}$, and the policy divergences are bounded as $\max_s D_{TV}(\pi_D(a|s)||\pi(a|s)) \leq \epsilon_\pi$. Then under a branched rollouts scheme with a branch length of $k$, the returns are bounded as:*

$$\eta[\pi] \geq \eta^{\text{branch}}[\pi] - 2r_{\max}\left[\frac{\gamma^{k+1}\epsilon_\pi}{(1-\gamma)^2} + \frac{\gamma^k\epsilon_\pi}{(1-\gamma)} + \frac{k}{1-\gamma}(\epsilon_{m'})\right]$$

*Proof.* As in the proof for Theorem A.1, the proof for this theorem requires adding and subtracting the correct reference quantity and applying the corresponding returns bound (Lemma B.4).

The choice of reference quantity is a branched rollout which executes the old policy $\pi_D$ under the true dynamics until the branch point, then executes the new policy $\pi$ under the true dynamics for $k$ steps. We denote the returns under this scheme as $\eta^{\pi_D, \pi}$. We can split the returns as follows:

$$\eta[\pi] - \eta^{\text{branch}} = \underbrace{\eta[\pi] - \eta^{\pi_D,\pi}}_{L_1} + \underbrace{\eta^{\pi_D,\pi} - \eta^{\text{branch}}}_{L_2}$$

We can bound both terms $L_1$ and $L_2$ using Lemma B.4.

$L_1$ accounts for the error from executing the old policy instead of the current policy. This term only suffers from error *before* the branch begins, and we can use Lemma B.4 $\epsilon_\pi^{\text{pre}} \leq \epsilon_\pi$ and all other errors set to 0. This implies:

$$|\eta[\pi] - \eta^{\pi_D,\pi}| \leq 2r_{\max} \left[ \frac{\gamma^{k+1}}{(1-\gamma)^2}\epsilon_\pi + \frac{\gamma^k}{1-\gamma}\epsilon_\pi \right]$$

$L_2$ incorporates model error *under the new policy* incurred after the branch. Again we use Lemma B.4, setting $\epsilon_m^{\text{post}} \leq \epsilon_m$ and all other errors set to 0. This implies:

$$|\eta[\pi] - \eta^{\pi_D,\pi}| \leq 2r_{\max} \left[ \frac{k}{1-\gamma}\epsilon_{m'} \right]$$

Adding $L_1$ and $L_2$ together completes the proof.

$\square$

The next bound is an analogue of Theorem A.2 except using model errors under the previous policy $\pi_D$ rather than the new policy $\pi$.

**Theorem A.3.** *Let the expected total variation between two the learned model is bounded at each timestep under the expectation of $\pi$ by $\max_t E_{s\sim\pi_{D,t}}[D_{TV}(p(s'|s,a)||\hat{p}(s'|s,a))] \leq \epsilon_m$, and the policy divergences are bounded as $\max_s D_{TV}(\pi_D(a|s)||\pi(a|s)) \leq \epsilon_\pi$. Then under a branched rollouts scheme with a branch length of $k$, the returns are bounded as:*

$$\eta[\pi] \geq \eta^{\text{branch}}[\pi] - 2r_{\max} \left[ \frac{\gamma^{k+1}\epsilon_\pi}{(1-\gamma)^2} + \frac{\gamma^k+2}{(1-\gamma)}\epsilon_\pi + \frac{k}{1-\gamma}(\epsilon_m + 2\epsilon_\pi) \right]$$

*Proof.* This proof is a short extension of the proof for Theorem A.2. The only modification is that we need to bound $L_2$ in terms of the model error under the $\pi_D$ rather than $\pi$.

Once again, we design a new reference rollout. We use a rollout that executes the old policy $\pi_D$ under the true dynamics until the branch point, then executes the *old* policy $\pi_D$ under the model for $k$ steps. We denote the returns under this scheme as $\eta^{\pi_D,\hat{\pi}_D}$. We can split $L_2$ as follows:

$$\eta^{\pi_D,\pi} - \eta^{\text{branch}} = \underbrace{\eta^{\pi_D,\pi} - \eta^{\pi_D,\hat{\pi}_D}}_{L_3} + \underbrace{\eta^{\pi_D,\hat{\pi}_D} - \eta^{\text{branch}}}_{L_4}$$

Once again, we bound both terms $L_3$ and $L_4$ using Lemma B.4.

The rollouts in $L_3$ differ in both model and policy after the branch. This can be bound using Lemma B.4 by setting $\epsilon_\pi^{\text{post}} = \epsilon_\pi$ and $\epsilon_m^{\text{post}} = \epsilon_m$. This results in:

$$|\eta^{\pi_D,\pi} - \eta^{\pi_D,\hat{\pi}_D}| \leq 2r_{\max} \left[ \frac{k}{1-\gamma}(\epsilon_m + \epsilon_\pi) + \frac{1}{1-\gamma}\epsilon_\pi \right]$$

The rollouts in $L_4$ differ only in the policy after the branch (as they both rollout under the model). This can be bound using Lemma B.4 by setting $\epsilon_\pi^{\text{post}} = \epsilon_\pi$ and $\epsilon_m^{\text{post}} = 0$. This results in:

$$|\eta^{\pi_D,\hat{\pi}_D} - \eta^{\text{branch}}| \leq 2r_{\max} \left[ \frac{k}{1-\gamma}(\epsilon_\pi) + \frac{1}{1-\gamma}\epsilon_\pi \right]$$

Adding $L_1$ from Theorem A.2 and $L_3$, $L_4$ above completes the proof.

$\square$

# B   Useful Lemmas

In this section, we provide proofs for various lemmas used in our bounds.

**Lemma B.1** (TVD of Joint Distributions). *Suppose we have two distributions $p_1(x,y) = p_1(x)p_1(y|x)$ and $p_2(x,y) = p_2(x)p_2(y|x)$. We can bound the total variation distance of the joint as:*

$$D_{TV}(p_1(x,y)||p_2(x,y)) \leq D_{TV}(p_1(x)||p_2(x)) + \max_x D_{TV}(p_1(y|x)||p_2(y|x))$$

*Alternatively, we have a tighter bound in terms of the expected TVD of the conditional:*

$$D_{TV}(p_1(x,y)||p_2(x,y)) \leq D_{TV}(p_1(x)||p_2(x)) + E_{x\sim p_1}[D_{TV}(p_1(y|x)||p_2(y|x))]$$

*Proof.*

$$
\begin{aligned}
D_{TV}(p_1(x,y)||p_2(x,y)) &= \frac{1}{2}\sum_{x,y}|p_1(x,y) - p_2(x,y)| \\
&= \frac{1}{2}\sum_{x,y}|p_1(x)p_1(y|x) - p_2(x)p_2(y|x)| \\
&= \frac{1}{2}\sum_{x,y}|p_1(x)p_1(y|x) - p_1(x)p_2(y|x) + (p_1(x) - p_2(x))p_2(y|x)| \\
&\leq \frac{1}{2}\sum_{x,y}p_1(x)|p_1(y|x) - p_2(y|x)| + |p_1(x) - p_2(x)|p_2(y|x) \\
&= \frac{1}{2}\sum_{x,y}p_1(x)|p_1(y|x) - p_2(y|x)| + \frac{1}{2}\sum_{x}|p_1(x) - p_2(x)| \\
&= E_{x\sim p_1}[D_{TV}(p_1(y|x)||p_2(y|x))] + D_{TV}(p_1(x)||p_2(x)) \\
&\leq \max_x D_{TV}(p_1(y|x)||p_2(y|x)) + D_{TV}(p_1(x)||p_2(x))
\end{aligned}
$$

$\square$

**Lemma B.2** (Markov chain TVD bound, time-varying). *Suppose the expected KL-divergence between two transition distributions is bounded as $\max_t E_{s\sim p_1^t(s)}D_{KL}(p_1(s'|s)||p_2(s'|s)) \leq \delta$, and the initial state distributions are the same $- p_1^{t=0}(s) = p_2^{t=0}(s)$. Then the distance in the state marginal is bounded as:*

$$D_{TV}(p_1^t(s)||p_2^t(s)) \leq t\delta$$

*Proof.* We begin by bounding the TVD in state-visitation at time $t$, which is denoted as $\epsilon_t = D_{TV}(p_1^t(s)||p_2^t(s))$.

$$
\begin{aligned}
|p_1^t(s) - p_2^t(s)| &= |\sum_{s'}p_1(s_t = s|s')p_1^{t-1}(s') - p_2(s_t = s|s')p_2^{t-1}(s')| \\
&\leq \sum_{s'}|p_1(s_t = s|s')p_1^{t-1}(s') - p_2(s_t = s|s')p_2^{t-1}(s')| \\
&= \sum_{s'}|p_1(s|s')p_1^{t-1}(s') - p_2(s|s')p_1^{t-1}(s') + p_2(s|s')p_1^{t-1}(s') - p_2(s|s')p_2^{t-1}(s')| \\
&\leq \sum_{s'}p_1^{t-1}(s')|p_1(s|s') - p_2(s|s')| + p_2(s|s')|p_1^{t-1}(s') - p_2^{t-1}(s')| \\
&= E_{s'\sim p_1^{t-1}}[|p_1(s|s') - p_2(s|s')|] + \sum_{s'}p(s|s')|p_1^{t-1}(s') - p_2^{t-1}(s')|
\end{aligned}
$$

$$\epsilon_t = D_{TV}(p_1^t(s)||p_2^t(s)) = \frac{1}{2}\sum_s |p_1^t(s) - p_2^t(s)|$$

$$= \frac{1}{2}\sum_s \left( E_{s'\sim p_1^{t-1}}[|p_1(s|s') - p_2(s|s')|] + \sum_{s'} p(s|s')|p_1^{t-1}(s') - p_2^{t-1}(s')| \right)$$

$$= \frac{1}{2}E_{s'\sim p_1^{t-1}}[\sum_s |p_1(s|s') - p_2(s|s')|] + D_{TV}(p_1^{t-1}(s')||p_2^{t-1}(s'))$$

$$= \delta_t + \epsilon_{t-1}$$

$$= \epsilon_0 + \sum_{i=0}^{t} \delta_t$$

$$= \sum_{i=0}^{t} \delta_t = t\delta$$

Where we have defined $\delta_t = \frac{1}{2}E_{s'\sim p_1^{t-1}}[\sum_s |p_1(s|s') - p_2(s|s')|]$, which we assume is upper bounded by $\delta$. Assuming we are not modeling the initial state distribution, we can set $\epsilon_0 = 0$. $\qquad\square$

**Lemma B.3** (Branched Returns bound). *Suppose the expected KL-divergence between two dynamics distributions is bounded as $\max_t E_{s\sim p_1^t(s)} D_{KL}(p_1(s', a|s)||p_2(s', a|s)) \leq \epsilon_m$, and $\max_s D_{TV}(\pi_1(a|s)||\pi_2(a|s)) \leq \epsilon_\pi$. Then the returns are bounded as:*

$$|\eta_1 - \eta_2| \leq \frac{2R\gamma(\epsilon_\pi + \epsilon_m)}{(1-\gamma)^2} + \frac{2R\epsilon_\pi}{1-\gamma}$$

*Proof.* Here, $\eta_1$ denotes returns of $\pi_1$ under dynamics $p_1(s'|s, a)$, and $\eta_2$ denotes returns of $\pi_2$ under dynamics $p_2(s'|s, a)$.

$$|\eta_1 - \eta_2| = |\sum_{s,a}(p_1(s,a) - p_2(s,a))r(s,a)|$$

$$= |\sum_{s,a}(\sum_t \gamma^t p_1^t(s,a) - p_2^t(s,a))r(s,a)|$$

$$= |\sum_t \sum_{s,a}\gamma^t(p_1^t(s,a) - p_2^t(s,a))r(s,a)|$$

$$\leq \sum_t \sum_{s,a}\gamma^t|p_1^t(s,a) - p_2^t(s,a)|r(s,a)$$

$$\leq r_{\max}\sum_t \sum_{s,a}\gamma^t|p_1^t(s,a) - p_2^t(s,a)|$$

We now apply Lemma B.2, using $\delta = \epsilon_m + \epsilon_\pi$ (via Lemma B.1) to get:

$$D_{TV}(p_1^t(s)||p_2^t(s)) \leq t(\epsilon_m + \epsilon_\pi)$$

And since we assume $\max_s D_{TV}(\pi_1(a|s)||\pi_2(a|s)) \leq \epsilon_\pi$, we get

$$D_{TV}(p_1^t(s,a)||p_2^t(s,a)) \leq t(\epsilon_m + \epsilon_\pi) + \epsilon_\pi$$

Thus, plugging this back in we get:

$$|\eta_1 - \eta_2| \leq r_{\max}\sum_t \sum_{s,a}\gamma^t|p_1^t(s,a) - p_2^t(s,a)|$$

$$\leq 2r_{\max}\sum_t \gamma^t t(\epsilon_m + \epsilon_\pi) + \epsilon_\pi$$

$$\leq 2r_{\max}(\frac{\gamma(\epsilon_\pi + \epsilon_m)}{(1-\gamma)^2} + \frac{\epsilon_\pi}{1-\gamma})$$

$\qquad\square$

467 **Lemma B.4** (Returns bound, branched rollout)**.** *Assume we run a branched rollout of length*
468 *k. Before the branch ("pre" branch), we assume that the dynamics distributions are*
469 *bounded as* $\max_t E_{s \sim p_1^t(s)} D_{KL}(p_1^{\text{pre}}(s',a|s)||p_2^{\text{pre}}(s',a|s)) \leq \epsilon_m^{\text{pre}}$ *and after the branch as*
470 $\max_t E_{s \sim p_1^t(s)} D_{KL}(p_1^{\text{post}}(s',a|s)||p_2^{\text{post}}(s',a|s)) \leq \epsilon_m^{\text{post}}$. *Likewise, the policy divergence is*
471 *bounded pre- and post- branch by* $\epsilon_\pi^{\text{pre}}$ *and* $\epsilon_\pi^{\text{post}}$, *repsectively. Then the K-step returns are bounded*
472 *as:*

$$|\eta_1 - \eta_2| \leq 2r_{\max} \left[ \frac{\gamma^{k+1}}{(1-\gamma)^2}(\epsilon_m^{\text{pre}} + \epsilon_\pi^{\text{pre}}) + \frac{k}{1-\gamma}(\epsilon_m^{\text{post}} + \epsilon_\pi^{\text{post}}) + \frac{\gamma^k}{1-\gamma}\epsilon_\pi^{\text{pre}} + \frac{1}{1-\gamma}\epsilon_\pi^{\text{post}} \right]$$

473 *Proof.* We begin by bounding state marginals at each timestep, similar to Lemma B.3. Recall that
474 Lemma B.2 implies that state marginal error at each timestep can be bounded by the state marginal
475 error at the previous timestep, plus the divergence at the current timestep.

476 Thus, letting $d_1(s,a)$ and $d_2(s,a)$ denote the state-action marginals, we can write:

477 For $t \leq k$:

$$TV d_1^t(s,a) d_2^t(s,a) \leq t(\epsilon_m^{\text{post}} + \epsilon_\pi^{\text{post}}) + \epsilon_\pi^{\text{post}} \leq k(\epsilon_m^{\text{post}} + \epsilon_\pi^{\text{post}}) + \epsilon_\pi^{\text{post}}$$

478 and for $t \geq k$:

$$TV d_1^t(s,a) d_2^t(s,a) \leq (t-k)(\epsilon_m^{\text{pre}} + \epsilon_\pi^{\text{pre}}) + k(\epsilon_m^{\text{post}} + \epsilon_\pi^{\text{post}}) + \epsilon_\pi^{\text{pre}} + \epsilon_\pi^{\text{post}}$$

479 We can now bound the difference in occupancy measures by averaging the state marginal error over
480 time, weighted by the discount:

$$D_{TV}(d_1(s,a)||d_2(s,a)) \leq (1-\gamma) \sum_{t=0}^{\infty} \gamma^t t D_{TV}(d_1^t(s,a)||d_2^t(s,a))$$

$$\leq (1-\gamma) \sum_{t=0}^{k} \gamma^t (k(\epsilon_m^{\text{post}} + \epsilon_\pi^{\text{post}}) + \epsilon_\pi^{\text{post}})$$

$$+ (1-\gamma) \sum_{t=k}^{\infty} \gamma^t (t-k)(\epsilon_m^{\text{pre}} + \epsilon_\pi^{\text{pre}}) + k(\epsilon_m^{\text{post}} + \epsilon_\pi^{\text{post}}) + \epsilon_\pi^{\text{pre}} + \epsilon_\pi^{\text{post}}$$

$$= k(\epsilon_m^{\text{post}} + \epsilon_\pi^{\text{post}} + \epsilon_\pi^{\text{post}}) + \frac{\gamma^{k+1}}{1-\gamma}(\epsilon_m^{\text{pre}} + \epsilon_\pi^{\text{pre}}) + \gamma^k \epsilon_\pi^{\text{pre}}$$

481 Multiplying this bound by $\frac{2r_{\max}}{1-\gamma}$ to convert the state-marginal bound into a returns bound completes
482 the proof. $\square$

## C Hyperparameter Settings

| | | HalfCheetah | Walker2d | Ant | Hopper |
|---|---|---|---|---|---|
| $N$ | epochs | 400 | | 300 | 125 |
| $E$ | environment steps per epoch | 1000 | | | |
| $M$ | model rollouts per policy update | 20 | | | |
| $B$ | ensemble size | 7 | | | |
| | network architecture | MLP with four hidden layers of size 200 | | | |
| $G$ | policy updates per epoch | 40000 | 20000 | | |
| $k$ | model horizon | 1 | | $1 \rightarrow 25$ over epochs $20 \rightarrow 100$ | $1 \rightarrow 15$ over epochs $20 \rightarrow 100$ |

Table 1: Hyperparameter settings for MBPO results shown in Figure 2. $x \rightarrow y$ over epochs $a \rightarrow b$ denotes a thresholded linear function, *i.e.* at epoch $e$, $f(e) = \min(\max(x + \frac{e-a}{b-a} \cdot (x - y), x), y)$