[Reviews · NeurIPS 2019]

Reviewer 1



The paper combines multiple known techniques in model-based RL, updates some of them, and achieves good empirical results. In addition, it includes a theoretical analysis of the improvement that can be achieved in the real world if using a model of the world to generate additional data to train the policy. The theoretical analysis brings rigour to the proposed approach, and is a welcome addition. In the end, the analysis is mainly used to justify the use of rollouts, and to give some understanding of how the length of the rollouts k affects the bounds. The conclusions that can be drawn from the analysis are somewhat limited compared to the size of the role the analysis plays in the paper, but I would still say it is useful for understanding the impact on performance of using internally simulated data for training the policy. I would encourage the authors to elaborate a bit further on the main intuitions behind the results of the theoretical analysis for readers who do not wish to go through all of the math. One contribution that seems to be important is to use rollouts not only from the initial state distribution, but from all steps of empirically collected data. I am surprised to hear that this has not been used extensively before, as it seems very logical to do. Nevertheless, I find studying this approach to be one of the most significant contributions of the paper. What I am struggling with is to understand what the significance are of the different parts of the contributions in the paper. The ablation studies are useful, but I am still puzzled by the large difference in performance to the comparison methods. Is it due to the k-step rollouts? How significant is the type of the probabilistic forward model? In general, it feels like all the methods in the comparison are mixed bags of different components that could in principle be combined quite freely. However, the evaluation is mainly done by comparing the entire bag with the other bags. This makes understanding the impact of different choices in the method difficult. Nevertheless, I think the authors have done a good job with the ablations, and have been able to alleviate this concern to a sufficient degree to qualify for a good paper. The empirical results presented in the paper are very good. I would be curious to see the method applied to more high-dimensional tasks than the ones presented (for example Humanoid - did you try it? My guess is that the exploitation of the learned model grows quite quickly with the dimensionality of the action space), although the benchmarks used already clearly show that the method is promising. The paper is clearly written, especially the related work section. I will unfortunately have to defer to the other reviewers to judge whether all relevant works have been cited. Overall, the paper presents a solid enough analysis of a novel enough method with good empirical results and extensive enough empirical data on the impact of design choices in the method. Edit after response: Thank you for the response, and especially for including the humanoid as a benchmark, it did make the paper stronger.

Reviewer 2



Summary: In model-based reinforcement learning, the traditional approaches often alternatively update between dynamic function from real samples and policy network from fictitious samples generated by this learned dynamic. The simplest way to generate these fictitious samples is to propagate the entire trajectories from scratch, starting from the same initial state. This paper argues that this propagation scheme is not so effective as the model errors are accumulated overtime; this explains why not much model-based RL algorithms are reported under long horizon tasks. To address this problem, they present another propagation approach; this time, the policy updating no longer relies on the entire long-horizon fictitious trajectories. Instead, a large number of short-horizon trajectories that are initiated under various visited state in real environment are collected to update the policy. In overall, the paper proposes an interesting idea to mitigate some issues in existing model-based RL. Its writing is very clear and easy to follow. The idea comes with good theoretical results and provides good insights. They are also linked well with the design of the proposed practical algorithm. Experimental results show an impressive results, where a proposed method considerably outperform almost all of the baseline. The paper’s idea is basically based on model ensemble learning, short rollout data and soft actor-critic. It initially looks incremental with little twist in the use of such short rollouts in policy learning, but turns out to be very efficient. In some sense, this algorithm can be viewed as ME-SAC. Therefore, I am concerned about the surprisingly poor performance of ME-TRPO (especially on Half Cheetah). I wonder if SAC uses Model-ensemble learning could also perform comparably with MBPO? However, the paper’s idea is backed by good theoretical results, so I think this is a strong work. Minor comments: - An elaborate about the working of k-branched rollout would be greater. - section titles should not have all initial capitals.

Reviewer 3



This work showed a way to bound the gap between performances of a policy on a learned model and the true model. It furthur showed how to leverage this bound to develop a practical model based algorithm. It is a very interesting work. The solution makes sense and the empirical evaluation is sufficient. weakness: this work relies on the bound in Eq. (1). However, this work doesn't discuss how tight is this bound. If the bound is not tight, how to improve it? If the bound is tight, why? I think the authors should provide more insights into the bound.

[Author Response · NeurIPS 2019]

We thank the reviewers for their detailed comments and positive evaluation of our work. The questions primarily concerned (1) evaluation in an environment with a larger action space, (2) performance of ME-TRPO, (3) analysis of the gap between MBPO and the baselines, and (4) discussion of the tightness of the theoretical bound.

**(R1) Code release.** Code for reproducing our experiments is now available on GitHub. To preserve anonymity, we do not link directly to the repository.

**(R1) Larger environment.** We provide results on the Humanoid environment, requested by R1, in Figure 1. We will add these to the final version.

**(R1) Intuitions for the theory.** We will expand the discussion in Section 4 to present a better intuition for the practical implications of the theory. The theory suggests that: (1) short-horizon rollouts may be beneficial in some settings; (2) incorporating the model can allow for larger policy changes while still achieving monotonic improvement, but only if the model generalizes well to changes in the policy – the worst-case generalization does not achieve this, but we empirically find that real models on MuJoCo benchmark tasks generalize substantially better.

**(R1, R2) Analysis of comparative performance.** R1 and R2 asked about the sources of improvement of our method over the baselines. Here we elaborate on our choice of ablations and how they address this question.

Figure 1: Results on Humanoid-v2. MBPO results are averaged over four seeds. The short rebuttal period did not allow for running all baselines to convergence, but we will add them to the final.

1. The 500-length rollout ablation in the paper's Figure 3 is the suggested ME-SAC baseline, as it uses the ensemble to generate model rollouts with lengths on the order of the task horizon for consumption by SAC. We conclude from this result that truncated rollouts are a primary source of the performance difference between our method and **ME-TRPO**.

2. To better understand the difference between using model data directly for training and for improved target value estimates (as done in **MVE** and **STEVE**), we implemented the value expansion technique on top of SAC to control for the underlying model-free algorithm. This comparison is found in the paper's Figure 3.

3. We do not include a separate ablation for **PETS** because the comparison between MBPO and PETS is already well-controlled. The model ensembles are the same in both methods, so the difference in performance is attributable to the different ways the model is used: planning by sampling from a fixed prior in PETS and policy optimization in MBPO.

**(R2) ME-TRPO baseline.** R2 raised concerns about the relatively poor performance of ME-TRPO on the full-length HalfCheetah. The ME-TRPO paper evaluates on modified tasks, with horizons of 100 or 200, making their reported results not representative of the standard benchmarks. Our results use the authors' code and are representative of the actual performance of the method. An independent benchmarking of model-based RL algorithms, released after the NeurIPS deadline, reported the same results from ME-TRPO on the full-length environments [Wang et al., 2019].

**(R2) Elaboration on branched rollouts.** The branching rollouts use the marginal distribution from a previous policy as an initial state distribution for truncated model rollouts. In practice, this amounts to sampling a state $s \sim \mathcal{D}$ from the environment replay buffer, rolling out under the model for at most $k$ steps using the current policy, and using these model predictions for policy optimization.

**(R3) Tightness of the bound.** Our bound is tightest in MDPs in which a single differing action or transition leads two trajectories to permanently diverge, as in the binary tree MDP in Figure 2. A crucial step in proving Theorem 1 is that if two agents select differing actions with $\epsilon$ probability, then their state marginals diverge by $\epsilon t$ in total variation (Lemma B2). In Figure 2, the amount of divergence is exactly $1 - (1 - \epsilon)^t$, which is close to $\epsilon t$ when $\epsilon$ is small. We are not aware of a way to create a tighter bound while still handling this pathological MDP. Similar proof techniques are used to analyze the TRPO and CPI algorithms.

### References

Tingwu Wang, Xuchan Bao, Ignasi Clavera, Jerrick Hoang, Yeming Wen, Eric Langlois, Shunshi Zhang, Guodong Zhang, Pieter Abbeel, and Jimmy Ba. Benchmarking model-based reinforcement learning. *arXiv preprint arXiv:1907.02057*, 2019.

Figure 2: An example MDP where the bound is presented in Theorem 1 is nearly tight.

[Meta-Review · NeurIPS 2019]

This paper brings together a number of techniques for MBRL, improving them as necessary, backed by a theoretical analysis as well as strong empirical results. Overall, a strong paper.